# Multidrug-Resistant and Extended-Spectrum β-Lactamase (ESBL) - Producing *Enterobacterales* Isolated from Carriage Samples among HIV Infected Women in Yaoundé, Cameroon

**DOI:** 10.3390/pathogens11050504

**Published:** 2022-04-24

**Authors:** Ravalona Jessica Zemtsa, Michel Noubom, Luria Leslie Founou, Brice Davy Dimani, Patrice Landry Koudoum, Aurelia Djeumako Mbossi, Charles Kouanfack, Raspail Carrel Founou

**Affiliations:** 1Faculty of Medicine and Pharmaceutical Sciences, University of Dschang, Dschang P.O. Box 96, Cameroon; jessicazemtsa@yahoo.com (R.J.Z.); noubommichel@yahoo.fr (M.N.); patricekoudoum@gmail.com (P.L.K.); charleskouanfack@yahoo.fr (C.K.); 2Reproductive, Maternal, Newborn and Child Health (ReMARCH) Research Unit, Research Institute of Centre of Expertise and Biological Diagnostic of Cameroon (CEDBCAM-RI), Yaoundé P.O. Box 67, Cameroon; luriafounou@gmail.com (L.L.F.); aureliadjeumako@gmail.com (A.D.M.); 3Dschang District Hospital, Foto, Dschang P.O. Box 43, Cameroon; 4Bioinformatics & Applied Machine Learning Research Unit, EDEN Biosciences Research Institute (EBRI), EDEN Foundation, Yaoundé P.O. Box 8242, Cameroon; 5Department of Biomedical Sciences, Higher Institute of Medical Technology, Nkolondom, Yaoundé P.O. Box 188, Cameroon; dimanydavy@gmail.com; 6Antimicrobial Resistance and Infectious Diseases, Research Institute of Centre of Expertise and Biological Diagnostic of Cameroon (CEDBCAM-RI), Yaoundé P.O. Box 8242, Cameroon; 7Central Hospital of Yaoundé, Yaoundé P.O. Box 47, Cameroon

**Keywords:** antibiotic resistance, Enterobacterales, HIV, MDR, ESBLs, Cameroon

## Abstract

The exacerbation of antimicrobial resistance (AMR) is a major public health threat worldwide. In sub-Saharan Africa, there is a scarcity of data regarding multidrug-resistant (resistance to at least one antibiotic of three or more families of antibiotics) as well as extended spectrum β-lactamase-producing *Enterobacterales* (ESBL-PE), isolated among clinical and asymptomatically healthy patients, especially in women living with HIV (WLHIV) despite their immunocompromised status. The overarching aim of this study was set to determine the prevalence and characterize genotypically multi-drug resistant *Enterobacterales* (MDR-E) and ESBL- PE isolated from vaginal swabs of WLHIV attending the Yaoundé Central Hospital, Yaoundé, Cameroon. A cross-sectional study was conducted among WLHIV during a four-month periods from 1 February to 31 May 2021. A total of 175 WLHIV, of childbearing age and under antiretroviral treatment were contacted. One hundred and twenty participants (120) were recruited and vaginal swabs were collected from them. After culture on Eosine-Methylen Blue (EMB) agar, the identification of *Enterobacterales* was performed using API 20E kit. A double-screening of ESBL-PE was performed using a combined disc diffusion method and ROSCO Diagnostica kits. An antibiotic susceptibility test was carried out by disc diffusion as per the Kirby-Bauer method and the β-lactamase resistance genes, *bla*_CTX-M_, *bla*_CTX-M-group1-2-9_, *bla*_TEM_ were molecularly characterized using a conventional Polymerase Chain Reaction (PCR). Overall, 30.83% (37/120) of the included WLHIV were colonized *with Enterobacterales* and the prevalence of vaginal carriage of MDR *Enterobacterales* among them was 62.16% (23/37). Among MDR-E isolates, the most prevalent species were *E. coli* (56.0%; 14/25) and *K. pneumoniae* (20.0%; 5/25). High rates of resistance to trimethoprim-sulfamethoxazole (96.0%; 24/25), amoxicillin-clavulanic acid (88.0%; 22/25) and gentamicin (72%; 18/25) were observed. The resistance mechanisms detected among these isolates were ESBL (48.0%; 12/25), ESBL+ porin loss (8.0%; 2/25), ESBL+AmpC (24%; 6/25), with *bla*_CTX-M_, *bla*_CTX-M-group-1,2,9_ being identified at 48.0% (12/25) for each of them and *bla*_TEM_ at 72.0% (18/25). Our findings confirm the high-prevalence of MDR as well as ESBL-PE isolated in WLHIV, and suggest that a real time monitoring system of antimicrobial resistant bacteria coupled with the reinforcement of infection prevention control (IPC) strategies are needed to sustainably contain these life-threatening pathogens especially in the most vulnerable populations.

## 1. Introduction

Antimicrobial resistance (AMR) remains a public health threat worldwide, especially in low- and middle-income countries (LMICs) where the limited hygiene and sanitation measures associated with numerous risk factors exacerbate this concern [1] (p. 10). *Enterobacterales* have been described as the main causative agents in numerous infectious diseases in healthcare settings and have been identified in the carriage and clinical samples of patients in communities, especially those with numerous co-morbidities [2] (p. 1874). Several mechanisms to escape the activities of antibiotics have been described among *Enterobacterales* and the most common is the production of an extended spectrum ß-lactamase as well as carbapenemases.

The World Health Organization (WHO) has ranked extended spectrum β-lactamase-producing *Enterobacterales* (ESBL-PE) as pathogens of critical priority in the research and development of new drugs [3] (p. 12). The emergence and global spread of multidrug resistant *Enterobacterales* (MDR-E), in hospital and community settings limits the use of therapeutic options, especially for people with co-morbidity conditions such as cancers and HIV [4] (p. 53–54). Several studies have highlighted the high prevalence of MDR-E and ESBL-PE in carriage samples of patients living with HIV (PLHIV) in Tanzania and Nigeria [5,6,7] (p. 611, 1542, 3). Furthermore, these studies postulated that HIV positive-patients are an important reservoir of ESBL-PE in Africa.

Moreover, there is a dearth of knowledge regarding the burden of MDR-E and ESBL-PE in people living with HIV, particularly in sub-Saharan Africa. A better understanding of the current epidemiology of these resistant pathogens in sub-Saharan Africa, the epicentre of the HIV pandemic (with around 67.0% of the world’s population living with HIV) and a region accounting for around 63.0% of new infections from women of childbearing age yearly will considerably improve health outcomes of this vulnerable population [8] (p. 3). In Cameroon, financial constraints preclude the active surveillance of AMR in general, MDR-E and ESBL-PE in particular. Neither the prevalence of MDR-E nor that of ESBL-PE colonizing or infecting women living with HIV (WLHIV) in communities and hospitals is known in the country. Such data could help to understand if antibiotic resistant *Enterobacterales* might be involved in infections in WLHIV, understand their transmission pathways and can help to determine their burden in this vulnerable population, thus serve as evidence to implement adequate prevention and containment measures of AMR in the country. This study therefore aims to phenotypically and genotypically characterize MDR and ESBL-PE isolated from vaginal swabs among WLHIV in Yaoundé, Cameroon.

## 2. Results

### 2.1. Sociodemographic and Clinical Characteristics

A total of 185 WLHIV were contacted during the sample collection period. Out of these, 120 (64.86%) agreed to participate and provided vaginal samples. The mean age of included women was 36.85 ± 7.22 years and median age was 37 years (min: 20 max: 50). Among the 120 included WLHIV, 37 (30.83%) were colonized by *Enterobacterales*, with 23 62.16% (23/37) being colonized by MDR-E. Of these women, 47.82% (11/23) also presented ESBL-PE. Among the 37 WLHIV colonized by an *Enterobacterales*, women between the ages of 35–39 (27.03%) were principally colonized followed by those between the ages of 40–44 (21.62%) and >45 (21.62%) years. Likewise, over 60.0% of WLHIV colonized by *Enterobacterales* had been to high school or had a high school degree (Table 1). Farmer (35.14%) and civil servant (21.62%) were the leading profession of the WLHIV colonized by an *Enterobacterales* isolate. Only five out of the 37 WLHIV were pregnant while all had a previous history of antibiotic treatment and an undetectable viral load of HIV. Among the WLHIV colonized by *Enterobacterales*, the majority of women aged between 40 and 44 years and those having been educated to a high school level were mainly colonized by MDR-E at respective percentages of 87.50% (7/8) and 66.67% (16/24). According to clinical parameters, 60% (3/5) of pregnant women were colonized by MDR-E and 20% (1/5) by ESBL-PE as shown in Table 1.

### 2.2. Distribution of Bacterial Species According to Multidrug Resistance and ESBL Production

Among the 37 women colonized by *Enterobacterales*, 11 women had two distinct colony morphotypes leading to 48 isolates. Out of these, 25/48 (52.08%) were MDR-E while 12/48 (25.0%) were concomitantly MDR-ESBL producers. *E. coli* was the leading MDR-E and ESBL-PE bacterial species at a respective prevalence of 56.0% (14/25) and 41.67% (5/12) whilst within the bacterial species, the highest frequency of MDR and ESBL production was observed among K. pneumoniae at a respective rate of 71.42% (5/7) and 57.14% (4/7). These results have high statistical significance (*p* < 0.0001) (Table 2). Interestingly, 23/48 (47.91%) isolates were neither MDR, ESBL nor MDR-ESBL producers and hence were considered as wild type (Table 2).

### 2.3. Antibiotic Resistance Patterns and Distribution of Resistances Mechanism among MDR-Enterobacterales

#### 2.3.1. Antibiotic Resistance among MDR-Enterobacterales Isolated

Table 3 shows an overall high level of resistance to amoxicillin-clavulanic acid (80.0%), piperacillin-tazobactam (88.0%) and trimethoprim-sulfamethoxazole (96.0%) among MDR-*Enterobacterales*. None of the isolates was resistant to imipenem. Specifically, *E. coli* was mainly resistant to trimethoprim- sulfamethoxazole (100%) while *K. pneumoniae* also exhibited high resistance to ceftriaxone and ceftazidime (100%).

MDR+ESBL positive *Enterobacterales* had a high level of resistance to all antibiotics used with the exception of imipenem. MDR+ESBL negative *Enterobacterales* had a low level of resistance to cefotaxime (15.38%), ceftazidime (7.69%), aztreonam (7.69%), ceftriaxone (38.46%) and netilmicin (30.76%) as shown in Figure 1.

After establishing the pattern of resistance, 7/25 (28.0%) of the MDR-E were resistant to 11 antibiotics and the dominant pattern was AUG-TPZ-CTX-CTR-CAZ- CFM-AZT-GEN-NET-COT-LEV (5/17; 29.41%). Among MDR-*E. coli* and MDR-ESBL positive *E. coli* the leading patterns were AUG-TPZ-GEN-CHL-LEV-COT (3/14; 21.42%) and AUG-TPZ-CTX-CTR-CAZ-AZT-GEN-NET-COT-LEV (2/5; 40%), respectively (Table 4).

#### 2.3.2. Resistance Mechanisms among MDR -Enterobacterales isolated in WLHIV at Yaoundé Central Hospital

Table 5 highlights that the main resistance mechanisms phenotypically detected among the isolates were ESBL (48.0%; 12/25) and OXA-48 (44.0%; 11/25).

### 2.4. Distribution of β-Lactamase Genes Mediating Resistance in MDR-Enterobacterales

Genes encoding for the production of β-lactamases detected in MDR-E isolates were *bla*_TEM_ at 72.0% (18/25) and *bla*_CTXM_ and *bla*_CTX-M-group-1, 2, 9_ each detected at 48.0% (12/25). These genes were predominant in *E. coli* (65%) and *K. pneumoniae* (20%) (Table 6).

Among the MDR-E isolates, which are concomitantly producers of ESBL, 83.33% (10/12) harboured the *bla*_TEM_, *bla*_CTX-M_ and *bla*_CTX-M-group-1, 2, 9_ genes. One strain carried only the *bla*_CTX-M_ and *bla*_CTX-M-goup-1, 2, 9_ genes, and one strain had only the *bla*_TEM_ (Table 7).

Figure 2 shows the 1.5% agarose gel electrolysis of conventional PCR amplification products of the *bla*_CTX-M_, *bla*_CTX-M-group-1,2,9_, *bla*_TEM_ genes detected from nine MDR-E isolates.

## 3. Discussion

The aim of this study was to characterize the phenotype and genotype of MDR-E and MDR-ESBL-Producing *Enterobacterales* isolated from vaginal swabs in WLHIV at the Yaoundé Central Hospital. Among the 120 included WLHIV, 23 (62.16%) were colonized by multi-drug resistant *Enterobacterales* (MDR-E) with women aged between 35–39 years (27.03%) representing the majority of those colonized. These results could be explained by the high prevalence of HIV/AIDS among women aged between 30 and 49 years in Cameroon [9] (p. 323). The majority of women colonized by MDR-E also had a high school education level (69.67%; 16/23), undetectable viral load (65.21%; 15/23) and had been under antiretroviral treatment for 1–5 years (34.7%; 8/23). Our results are higher than those obtained in Gondar, Ethiopia, in a study of the faecal carriage of ESBL-PE in children living with HIV, where children that had an undetectable viral load (45.16%) presented the highest number of ESBL-PE in faecal samples. The difference observed may be due to the difference of population study and sample types [10] (p. 3).

*E. coli* (56.0%; 14/25) and *K. pneumoniae* (20.0%; 5/25) were the most prevalent species among the MDR-E isolates. These rates are similar to those obtained in an Ethiopian study assessing *Enterobacterales* in clinical samples where *E. coli* accounted for 51.54% (150/291) of MDR-E isolates [11] (p. 4–5). Extended spectrum β-lactamase producing *Enterobacterales* (ESBL-PE) isolates were all MDR in our study, representing 48% (n = 12/25) of MDR-E isolates. We surmise that this might be due to the co-existence of genes encoding for these ß-lactamases along with other resistance genes encoding for resistance to other antibiotic families, which has been regularly reported among ESBL-PE in the literature as they are known to be associated with mobile genetic elements such as plasmids [12] (p. 1175). *E. coli* (41.16%) and *K. pneumoniae* (33.33%) were the leading ESBL-PE species in our study. These results are comparable to those obtained in a study conducted in Togo, where *E. coli* (63.64%) and *K. pneumoniae* (27.27%) were the most common ESBL-PE species isolated among vaginal swabs in clinical samples [12] (p. 1168). The presence of ESBL-PE in the vaginal microbiome could be associated with (i) the weakness of the immune system of WLHIV, (ii) the development of resistance mechanisms due to prolonged ARV treatment or prophylactic antibiotic use, or (iii) limited hygienic measures [13] (p. 10).

Antibiotic susceptibility testing showed that the all MDR-E isolates displayed high resistance to trimethoprim-sulphamethoxazole (96.0%), piperacillin-tazobactam (88.0%), amoxicillin-clavulanic acid (80.0%) and chloramphenicol (68.0%). The high resistance rate to these antibiotics could be explained by the fact that they are readily available over-the-counter in Cameroon and are thus used for self-medication. In addition, trimethoprim-sulphamethoxazole is recommended to HIV-positive adults for prophylaxis against opportunistic infections [13] (p. 9–10). This apparent unwarranted consumption of antibiotics has likely triggered the emergence of MDR in the human microbiome in Cameroon. The elevated rate of multi-drug resistance within *E. coli* and *K. pneumoniae* indicates that the effectiveness of antibiotics commonly used for bacterial infections among WLHIV is therefore threatened. This is even more worrying given that five of the WLHIV colonized by *Enterobacterales* were pregnant and three were colonized by MDR-E or MDR-ESBL-PE. Consequently, it is plausible that these pregnant women could not only suffer from a difficult-to-threat infection, but could also transmit the MDR-E or MDR-ESBL-PE isolates to their neonates either in utero or during delivery, thereby threatening the health of their progenies.

Our data confirm high rates of concomitant resistance to several antibiotics especially among MDR-ESBL positive compared to MDR-ESBL negative-*Enterobacterales*. This indicates that MDR-ESBL-PE are significantly more resistant to various antibiotic classes than those MDR- ESBL negative *Enterobacterales*. Interestingly, none of the MDR-ESBL positive and MDR-ESBL negative *Enterobacterales* were resistant to imipenem and up to twelve *Enterobacterales* were wild type with resistance to a maximum of two antibiotics. This result is highly relevant as it reveals that it is the time to act to preserve the efficacy of antibiotics for future generations. It further suggests that appropriate prevention and containment measures are urgently needed to sustainably curb AMR.

Interestingly, despite the high susceptibility of imipenem, *bla*_OXA-48_ (44%; 11/25) and *bla*_KPC_ (4%; 1/25) have been phenotypically detected among the MDR-E isolates. These enzymes are among the main carbapenemase found in *Enterobacterales*. These data are worrying because ESBL-PE infections are treated as a last resort with carbapenems. Infections caused by carbapenemase-producing *Enterobacterales* (CPE) are generally associated with a very poor prognosis and high mortality, especially in immunocompromised patients [14] (p. 2). AmpC production has also been detected between MDR-E isolates at a rate of 20% (5/25). This enzyme is responsible for resistance to cephalosporin, penicillin and beta-lactamase inhibitors [15] (p. 61).

In this study, *bla*_TEM,_
*bla*_CTX m_ and *bla*_CTX-M-group-1, 2, 9_ were detected at respective percentages of 72.0% (18/25), 48.0% (12/25) and 48.0% (12/25). Between MDR-ESBL-PE isolates, these bla genes were detected at similar percentages of 91.66% (11/12). These results are different from those obtained in Nepal among ESBL-PE isolated from HIV patients where the rates were 23.9%, and 86.9% for *bla*_TEM,_ and *bla*_CTX-M_, respectively [16] (p. 8). The high prevalence of *bla*_TEM,_ among our strains could be explained by the regular contact of WLHIV with hospital and healthcare professionals to monitor their HIV/AIDS infection [17] (p. 7).

The *bla*_CTX-M_ were the leading genes among MDR-ESBL-PE isolates with a 91.66% (11/12) prevalence. This could be due to the irrational consumption of ß-lactam antibiotics, especially cefotaxime, among WLHIV in Cameroon. These genes were also predominant among ESBL-PE isolated from HIV patients in Zimbabwe [5] (p. 611) and in Tanzania [6] (p. 1542). This is clearly in line with several reports which showed that CTX-M is the leading ESBL gene across the world. Following the detection of CTX-M-_group-1, 2, 9_ among MDR-E, we postulate the presence of *bla*_CTX-M-15_ among these isolates. Indeed, *bla*_CTX-M-15_ is part of group 1 of CTX-M, which is the most widely distributed across the world, and some studies have reported their presence in Cameroon [18,19] (p. 38, 5347). The presence of CTX-M among WLHIV is a danger. Not only because it encodes for ESBL, but also because it is carried by plasmids and could therefore horizontally spread within and between species as well as across hosts [16] (p. 12). Furthermore, as these women are of childbearing age, during an eventual pregnancy, vertical transmission of these MDR-E carrying *bla_CTX-M_* genes could occur, likely complicating the therapeutic choice in case of an infection of their newborns.

## 4. Materials and Methods

### 4.1. Study Settings

This study was conducted over a period of four months from 1 February to 31 May 2021 at one of biggest tertiary health care centre in Yaoundé, the Yaoundé Central Hospital (YCH). This healthcare structure provides patients with medical and paramedical teams specialised in various medical fields. It has a total of 650 beds, 70 doctors, 408 paramedical staff and 114 administrative staff and state agents. The day hospital, which is a fully fledged service of the YCH, is a pilot program for the fight against HIV/AIDS. It coordinates its activities with approximately 12,000 people living with HIV and is designated as an aggregated centre with a therapeutic committee, an ethics committee, a laboratory, and a pharmacy to ensure the distribution of anti-retroviral (ARVs).

### 4.2. Study Population

Our study included WLHIV, of childbearing age and under ARVs treatment at the YCH. All eligible women attending the day hospital were asked to participate in the study. Those who provided their informed consent to participate in the study were recruited. Socio-demographic and clinical information was recorded from pre-designed questionnaires on EPI INFO (version 7.7.3). Following this step, information was codified in order to ensure confidentiality.

### 4.3. Sample Collection

Vaginal swabs were collected aseptically in an appropriate sampling room. After an explanation of the sampling procedure, the patient was placed on a gynaecological bed and after proper speculum insertion the samples were collected with a sterile cotton swab.

### 4.4. Laboratory Analyses

The samples were immediately sent to the microbiology unit of the YCH laboratory for microbiological analysis and thereafter the isolates were molecularly characterized in the reproductive, maternal, neonatal and child health research unit of the Centre of Expertise and Biological Diagnostic of Cameroon (CEDBCAM).

#### 4.4.1. Culture and Identification

Vaginal swab samples were directly plated on to Eosin Methylene Blue (EMB) agar. Culture media were thereafter incubated overnight at 37 °C for 18–24 h. Growing colonies were primarily identified based on their morphological characteristics, then subjected to Gram-staining and an oxidase test which was conducted to ascertain the *Enterobacteriaceae* to the family level. The API 20E kit (BioMérieux, Marcy l’Etoile, France) was used to biochemically characterize *Enterobacterales* species according to the manufacturer’s instructions.

#### 4.4.2. Antimicrobial Susceptibility Testing (AST)

The AST of isolated *Enterobacterales* was assessed using the Kirby-Bauer disc diffusion method. Briefly, Mueller-Hinton agar was swabbed with 0.5 McFarland bacterial inoculum. Subsequently, a panel of 12 antibiotics discs of five different families was tested. These include amoxicillin-clavulanic acid (20–10 µg), piperacillin-tazobactam (30–6 µg), cefotaxime (30 µg), ceftriaxone (30 µg), ceftazidime (30 µg), netilmicin (10 µg), gentamicin (30 µg), imipenem (10 µg), levofloxacin (5 µg), trimethoprim-sulfamethoxazole (23.75–1.25 µg), aztreonam (30 µg), and chloramphenicol (10 µg). The plates were then incubated at 37 °C for 18–24 h. The different diameters of the inhibition zones were measured and interpreted as susceptible (S), intermediate (I) or resistant (R) according to the criteria defined by Antibiogram Committee French Society of Microbiology [20] (p. 38–47).

#### 4.4.3. Screening and Confirmation for ESBL- and Carbapenemase -Producing Enterobacterales

Phenotypic screening of ESBLs production among our isolates was performed by looking for champagne cork or funnel shaped synergies according to the CA-SFM recommendation [20] (p. 40). This first test was conducted using a combination of amoxicillin-clavulanic acid and cefotaxime, ceftriaxone and aztreonam to screen for ESBL production and amoxicillin-clavulanic acid and imipenem to screen for carbapenemase production. A second screening was performed using ROSCO Diagnostica kit according to the manufacturer’s instructions to ascertain ESBLs, ESBL-porin loss, AmpC, KPC, MBL and OXA-48 in our isolates.

#### 4.4.4. Genomic Extraction

The genomic DNA of the MDR-E isolates was obtained using a modified extraction by the boiling method as described previously [21] (p. 5347). Briefly, one to two pure MDR-E colonies were suspended in 400 μL of Tris-EDTA (10 mM Tris, 0.1 mM EDTA) and then vortexed for five seconds. The suspension was then incubated for 25 min at 95 °C in a dry bath digital (MIULab DKT200-1, Lasec International Ltd., Johannesburg, South Africa). After incubation, the suspension was centrifuged for 5 min at 9500 rpm and then, 150 μL of the supernatant containing DNA was subsequently transferred to a new Eppendorf tube then stored at –40 °C.

#### 4.4.5. Conventional Polymerase Chain Reaction

The detection of the *bla*_TEM_, *bla*_CTX-M_ and *bla*_CTX-M-group-1,2,9_, genes Among the MDR-E isolates was performed by conventional PCR in thermal cycler BIO-RAD T100 (Bio-Rad Laboratories, Marnes-la-Coquette, France). The reaction occurred in a 25 μL reaction mix consisting of 9 μL of 2x *Taq* green polymerase master mix (New England Biolabs, Ipswich, MA, USA), 12.5 μL of nuclease-free water, 0.25 μL of each reverse and forward primer [10 µM] and 3 μL of DNA. The amplification steps were as follows: initial denaturation (94 °C for 30 s), 30 cycles of denaturation at 94 °C for 4 s annealing for 40 s (TEM) and 45 s (CTX m universal and CTX-M-group-1,2,9), elongation at 72 °C for 50 s (CTX m universal and CTX-M-group-1,2,9) and 60 s (TEM) and final elongation at 72 °C for 5 min (TEM) and 6 min (CTX m universal and CTX-M-group-1,2,9). Annealing temperatures and primer sequences are showed in Table 8.

#### 4.4.6. DNA Electrophoresis and Visualization

After amplification, DNA electrophoresis was performed on agarose gel of 1.5% (*v/v*) that was run at 90 V for 45 min along with a 100 bp molecular ladder (New England Biolabs, MA, USA). The gel was then stained in ethidium bromide solution (0.5 µg/mL) and PCR products were visualised under UV light using a gel documentation system G-BOX chemi XL (Syngene, Cambridge, UK).

### 4.5. Data Analysis

Data analysis was performed using R software (version 4.1.0) and RStudio (version 2021.09.0). Proportions were compared using the Fischer exact test and two-sample t-test as appropriate. A participant was considered positive for MDR-E or ESBL-PE when at least one colony displaying the multidrug resistant or ESBL phenotype was detected. Results were considered statistically significant at a *p-value* < 0.05.

## 5. Conclusions

Our study revealed a high prevalence of MDR-E and MDR-ESBL-producing-*Enterobacterales* in the vaginal flora of WLHIV at Yaoundé Central Hospital. *E. coli* and *K. pneumoniae* were the leading resistant bacterial species while *bla*_TEM_ and *bla*_CTX-M_ were detected at elevated rates. These MDR-E, colonizing WLHIV, threaten not only WLHIV because they could be responsible for serious difficult-to-treat bacterial infections among these already vulnerable women, but also could compromise their reproductive health and health of their future progenies in case of intrapartum or neonatal (early or late onset) infections. These results evidence the need to implement stringent infection prevention and control measures, and to develop adequate strategies to monitor and limit the dissemination of MDR-*Enterobacterales* in this population.

## Figures and Tables

**Figure 1 pathogens-11-00504-f001:**
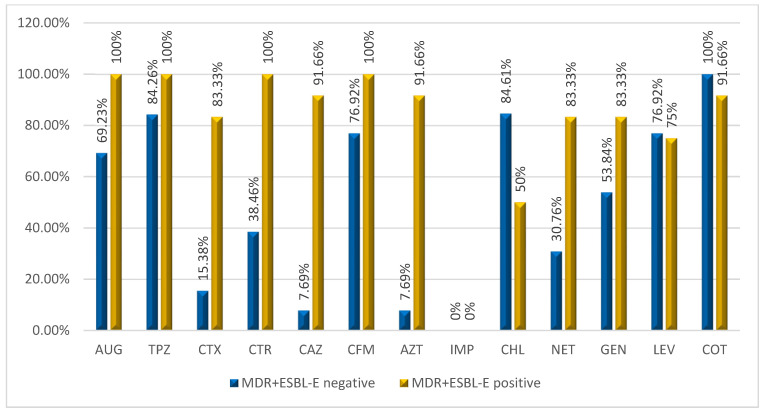
Resistance levels of MDR+ESBL positive and MDR+ESBL negative *Enterobacterales.*
**AUG**: Amoxicillin-clavulanic acid, **TPZ**: Piperacillin–tazobactam, **CTX:** Cefotaxime, **CTR**: Ceftriaxone, **CAZ**: Ceftazidime, **AZT**: Aztreonam, **IMP:** Imipenem, **GEN**: Gentamicin, **NET**: Netilmicin, **TMP/SXT**: Trimethoprim-sulfamethoxazole, **CHL**: Chloramphenicol, **LEV**: Levofloxacin.

**Figure 2 pathogens-11-00504-f002:**
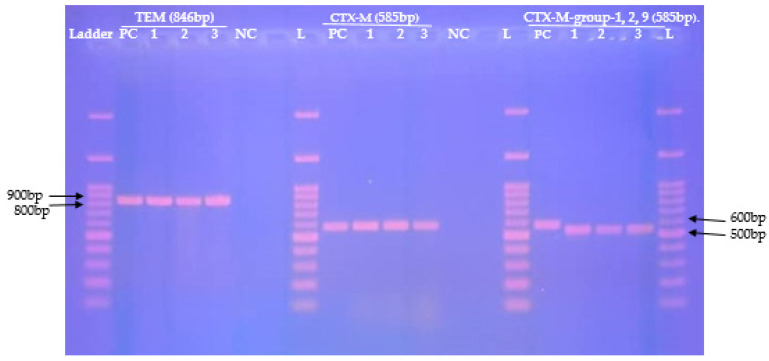
Agarose gel electrophoresis of amplified *bla* genes (*bla*_TEM_, *bla*_CTX-M_, *bla*_CTX-M-group-1, 2, 9_) from nine (09) MDR-***E*** isolated from vaginal swabs among HIV infected patients. **L**: 100 bp DNA ladder, **NC**: negative control, **PC**: positive control.

**Table 1 pathogens-11-00504-t001:** Summary of socio-demographics characteristics of women living with HIV according to MDR- and ESBL-positive status.

Variables	*Enterobacterales* Positive n (%)	MDR-*Enterobacterales* Positive n (%)	MDR-ESBL-*Enterobacterales* Positive n (%)
**Overall**	37 (31.0)	23 (62.0)	11 (47.82)
**Age**			
**Median [IQR ^#^]**	39 [34.0; 44.0]	36 [31.0; 41.0]	36 [31.0; 41.0]
**Mean [std *]**	38.7 [7.3]	36 [7.1]	36 [7.1]
**Age Group**			
**20** **–** **24**	2 (5.41)	1 (50.0)	1 (50.0)
**25** **–** **29**	2 (5.41)	1 (50.0)	1 (50.0)
**30** **–** **34**	7 (18.92)	2 (28.0)	2 (28.57)
**35** **–** **39**	10 (27.03)	6 (60.0)	2 (20.0)
**40** **–** **44**	8 (21.62)	7 (87.50)	1 (12.50)
**>45**	8 (21.62)	6 (75.0)	3 (37.50)
**Education**			
**Illiterate**	0	0	0
**Primary school**	4 (10.81)	3 (75.0)	2 (50.0)
**High school**	24 (64.86)	16 (66.67)	6 (25.0)
**University**	9 (24.32)	4 (44.44)	2 (22.22)
**Profession**			
**Seller**	6 (16.22)	5 (83.33)	2 (33.33)
**Student**	2 (5.41)	1 (50.0)	1 (50.0)
**Housewife**	7 (18.92)	3 (42.86)	2 (28.57)
**Civil servant**	8 (21.62)	6 (75.0)	3 (37.50)
**Farmer**	13 (35.14)	7 (53.85)	2 (15.38)
**Healthcare worker**	1 (2.70)	1 (100)	0
**Pregnancy Status**			
**No**	32 (86.49)	20 (62.50)	9 (28.12)
**Yes**	5 (13.51)	3 (60.0)	1 (20.0)
**Vaginal Hygiene**			
**No**	7 (18.92)	4 (57.14)	1 (14.29)
**Yes**	30 (81.08)	19 (63.33)	9 (30.0)
**Previous antibiotic**			
**No**	0	0	0
**Yes**	37 (100)	23 (62.16)	10 (27.03)
**Viral Load**			
**Unknown**	5 (13.51)	3 (60.0)	3 (60.0)
**<1000 copies**	1 (2.70)	1 (100)	0
**>1000 copies**	6 (16.22)	4 (66.67)	3 (50.0)
**Undetectable**	25 (67.57)	15 (60.0)	4 (16.0)
**Period under ARV Treatment**			
**1** **–** **5 years**	15 (40.54)	8 (53.33)	2 (13.33)
**6** **–** **10 years**	9 (24.32)	7 (77.78)	4 (44.44)
**10** **–** **15 years**	11 (29.73)	7 (63.64)	3 (27.27)
**16** **–** **20 years**	2 (5.41)	1 (50.0)	1 (50.0)
**>20 years**	0	0	0

* std: standard deviation, ^#^ IQR: Interquartile range, ARV: antiretroviral.

**Table 2 pathogens-11-00504-t002:** Distribution of ESBL producers among MDR and non-MDR *Enterobacterales* isolated in WLHIV.

*Bacterial Species (n)*	*MDR-ESBL Negative* *n (%)*	*MDR+ESBL Negative* *n (%)*	*p-Value*	*MDR+ESBL* *Positive n (%)*	*p-Value*
** *Escherichia coli (29)* **	15 (51.72)	9 (31.03)	<0.0001	5 (17.24)	<0.0001
** *Klebsiella pneumoniae (7)* **	2 (28.57)	1 (14.28)		4 (57.14)	
** *Klebsiella oxytoca (2)* **	0	1 (50.0)		1 (50.0)	
** *C. freundii (1)* **	1 (100)	0		0	
** *Enterobacter aerogenes (1)* **	0	1 (100)		0 (0)	
** *Enterobacter sakazaki (1)* **	1 (100)	0		0	
** *Enterobacter gergoviae (2)* **	1 (50.0)	0		1 (50.0)	
** *Serratia fonticola (2)* **	1 (50.0)	0 (0)		1 (50.0)	
** *Serratia marcescens (2)* **	2 (100)	0		0	
** *Raoultella ornithynolitica (1)* **	0	1 (100)		0 (0)	
**Overall, N = 48**	23 (48.0)	13 (27.0)		12 (25.0)	

**Table 3 pathogens-11-00504-t003:** Distribution of antibiotic resistance among all MDR-*Enterobacterales* isolated from WLHIV at the Yaoundé Central Hospital.

**Bacterial Species**	**ß-Lactams n (%)**						
	**AUG**	**TPZ**	**CTX**	**CTR**	**CAZ**	**AZT**	**IMP**
**Overall (n = 25)**	22 (88.0)	24 (96.0)	11 (44.0)	14 (56.0)	14 (56.0)	12 (48.0)	0
** *E. coli (n = 14)* **	12 (85.7)	14 (100)	5 (35.71)	6 (42.85)	6 (42.85)	5 (35.71)	0
** *K. pneumoniae (n = 5)* **	4 (80.0)	5 (100)	4 (80.0)	5 (100)	5 (100)	4 (80.0)	0
** *E. gergoviae (n = 1)* **	1 (100)	1 (100)	0	1 (100)	1 (100)	1 (100)	0
** *E. aerogenes (n = 1)* **	1 (100)	1 (100)	0	0	0	0	0
** *S. fonticola (n = 1)* **	1 (100)	1 (100)	1 (100)	1 (100)	1 (100)	1 (100)	0
** *K. oxytoca (n = 2)* **	2 (100)	2 (100)	1 (50.0)	1 (50.0)	1 (50.0)	1 (50.0)	0
** *R. ornithynolitica (n = 1)* **	1 (100)	0	0	0	0	0	0
**Bacterial Species**	**Non-ß-Lactam Antibiotics n (%)**						
	**GEN**	**NET**	**TMP/SXT**	**CHL**	**LEV**		
**Overall (n = 25)**	18 (72.0)	14 (56.0)	24 (96.0)	18 (72.0)	20 (80.0)		
** *E. coli (n = 14)* **	10 (71.42)	6 (42.85)	14 (100)	11 (78.57)	10 (71.42)		
** *K. pneumoniae (n = 5)* **	4 (80.0)	3 (60.0)	4 (80.0)	2 (50.0)	4 (80.0)		
** *E. gergoviae (n = 1)* **	1 (100)	1 (100)	1 (100)	1 (100)	1 (100)		
** *E. aerogenes (n = 1)* **	1 (100)	1 (100)	1 (100)	1 (100)	1 (100)		
** *S. fonticola (n = 1)* **	1 (100)	1 (100)	1 (100)	0	1 (100)		
** *K. oxytoca (n = 2)* **	1 (50.0)	1 (50.0)	2 (100)	2 (100)	2 (100)		
** *R. ornithynolitica (n = 1)* **	0	1 (100)	1 (100)	1 (100)	1 (100)		

**AUG**: Amoxicillin-clavulanic acid, **TPZ**: Piperacillin-tazobactam, **CTX**: Cefotaxime, **CTR**: Ceftriaxone, **CAZ**: Ceftazidime, **AZT**: Aztreonam, **IMP**: Imipenem, **GEN**: Gentamicin, **NET**: Netilmicin, **TMP/SXT**: Trimethoprim-sulfamethoxazole, **CHL**: Chloramphenicol, **LEV**: Levofloxacin.

**Table 4 pathogens-11-00504-t004:** Antimicrobial resistance profile of the 25 MDR *Enterobacterales* isolated among WLHIV.

Phenotypic Profile	Number of Family of Antibiotics	Number of Antibiotics	Number of Isolates (%)
TPZ-CHL-LEV-COT	4	4	2 (8.33)
AUG-TPZ-NET-CHL-COT	4	5	1 (4.16)
AUG-TPZ-GEN-CHL-COT			2 (8.33)
AUG-TPZ-CHL-LEV-COT			1 (4.16)
AUG-TPZ-GEN-CHL-LEV-COT	5	6	2 (8.33)
AUG-TPZ-AZT-GEN-CHL-COT	5		1 (416)
AUG-TPZ-NET-GEN-CHL-COT	4		1 (4.16)
AUG-CTR-NET-CHL-LEV-COT	6		1 (4.16)
CTX-CTR-CAZ-ATM-CHL-COT	4	6	1 (4.16)
AUG-TPZ-CTR-CAZ-AZT-NET-COT-LEV	6	8	1 (4.16)
AUG-TPZ-CTR-CAZ-AZT-COT-CHL-LEV	6		1 (4.16)
AUG-TPZ-CTX-CTR-CAZ-GEN-NET-COT-LEV	5	9	1 (4.16)
AUG-TPZ-CTX-CTR-CAZ-AZT-GEN-CHL-LEV	6	10	2 (8.33)
AUG-TPZ-CTX-CTR-CAZ-AZT-GEN-NET-COT	6		1 (4.16)
AUG-TPZ-CTX-CTR-CAZ-AZT-GEN-NET-COT-LEV	6	11	5 (20.83)
AUG-TPZ-CTR-CAZ-AZT-GEN-NET-COT-CHL-LEV	7		1 (4.16)
AUG-TPZ-CTX-CTR-CAZ-AZT-GEN-NET-COT-CHL-LEV	7	12	1 (4.16)

**AUG**: Amoxicillin-clavulanic acid, **TPZ**: Piperacillin-tazobactam, **CTX**: Cefotaxime, **CTR**: Ceftriaxone, **CAZ**: Ceftazidime, **AZT**: Aztreonam, **IMP**: Imipenem, **GEN**: Gentamicin, **NET**: Netilmicin, **TMP/SXT**: Trimethoprim-sulfamethoxazole, **CHL**: Chloramphenicol, **LEV**: Levofloxacin.

**Table 5 pathogens-11-00504-t005:** Distribution of resistance mechanisms among MDR -*Enterobacterales* isolated in WLHIV at Yaoundé Central Hospital.

Bacterial Species	N (%)	Resistance Mechanism					
		ESBL	ESBL+AmpC	ESBL + Loss of Porin	AmpC	OXA-48	KPC
*E. coli*	14	5 (35.71)	1 (6.67)	1 (6.67)	2 (13.33)	3 (20.0)	1 (6.67)
*K. pneumoniae*	5	4 (80.0)	3 (60.0)	0 (0)	3 (60.0)	4 (80.0)	0 (0)
*E. gergoviae*	1	1 (100)	1 (100)	0 (0)	0 (0)	1 (100)	0 (0)
*E. aerogenes*	1	0 (0)	1 (100)	0 (0)	0 (0)	1 (100)	0 (0)
*S. fonticola*	1	1 (100)	0 (0)	0 (0)	0 (0)	1 (100)	0 (0)
*K. oxytoca*	1	1 (100)	0 (0)	1 (100)	0 (0)	1 (100)	0 (0)
*R. ornithynolitica*	1	0 (0)	0 (0)	0 (0)	0 (0)	0 (0)	0 (0)
Overall n (%)	25	12 (48.0)	6 (24.0)	2 (8.0)	5 (20.0)	11 (44.0)	1 (4.0)

**Table 6 pathogens-11-00504-t006:** Distribution of β-lactamase genes between MDR-*Enterobacterales.*

MDR-E Species	*MDR+ESBL Positive* n (%)	β-Lactamases Genes		
		*bla*_CTX-M_ n (%)	*bla*_CTX-M-group-1,2,9_ n (%)	*bla*_TEM_ n (%)
*Overall, n = 25*	12 (48.0)	12 (48.0)	12 (48.0)	18 (72.0)
*E. coli*; *(n = 14)*	5 (35.71)	6(42.85)	6(42.85)	11 (78.57)
*K. pneumoniae*; *(n = 5)*	4 (80.0)	4 (80.0)	4 (80.0)	4 (80.0)
*K. oxytoca*; *(n = 2)*	1 (50.0)	1 (50.0)	1 (50.0)	1 (50.0)
*E. aerogenes*; *(n = 1)*	0 (0)	0 (0)	0 (0)	1 (100)
*E. gergoviae*; *(**n = 1)*	1 (100)	0 (0)	0 (0)	1 (100)
*S. fonticola*; *(**n = 1)*	1 (100)	1 (100)	1 (100)	1 (100)

**Table 7 pathogens-11-00504-t007:** Distribution of ß-lactamase genes among MDR-Enterobacterales according to their phenotypic characterization.

MDR-E Species (n)	Phenotypic Screennig of ESBL-PE	*bla* Resistances Genes	N (%)
*E. coli (n = 14)*	Positive	*bla*_CTX-M_, *bla*_CTX-M-group-1,2,9_, *bla*_TEM_	4 (28.57%)
		*bla*_CTX-M_, *bla*_CTX-M-group-1,2,9_	1 (7.14%)
	Negative	*bla* _TEM_	6 (42.8%)
	Negative	*bla*_CTX-M_, *bla*_CTX-M-group-1,2,9_, *bla*_TEM_	1 (7.14%)
*K. pneumoniae (n = 5)*	Positive	*bla*_CTX-M_, *bla*_CTX-M-group-1,2,9_, *bla*_TEM_	4 (80.0%)
*K. oxytoca (n = 2)*	Positive	*bla*_CTX-M_, *bla*_CTX-M-group-1,2,9_	1 (50.0%)
*E. aerogenes (n = 1)*	Negative	*bla* _TEM_	1 (100%)
*E. gergoviae (n = 1)*	Positive	*bla* _TEM_	1 (100%)
*S. fonticola (n = 1)*	Positive	*bla*_CTX-M_, *bla*_CTX-M-group-1,2,9_, *bla*_TEM_	1 (100%)

**Table 8 pathogens-11-00504-t008:** Characteristics of PCR primers.

*Bla* Targeted	Primer Name	Sequence (5′-3′)	Amplicon Size (bp)	Annealing Temperatures	References
TEM	TEMF TEMR	CATTTCCGTGTCGCCCTTATTC CCAATGCTTAATCAGTGAGGC	846 pb	49.2 °C	[22] (p. 712)
CTX-M	CTX-MU1 CTX-MU2	CGATGTGCAGTACCAGTAA TTAGTGACCAGAATCAGCGG	585 pb	46.6 °C	[23] (p. 1319)
CTX-M-group-1,2,9	CTX-MA1 CTX-MA2	SCSATGTGCAGYACCAGTAA CCGCRATATGKTTGGTGGTG	585 pb	56.2 °C	[24] (p. 29)

## Data Availability

Not applicable here.

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
