# Peer review of "Multidrug-Resistant and Extended-Spectrum β-Lactamase (ESBL) - Producing Enterobacterales Isolated from Carriage Samples among HIV Infected Women in Yaoundé, Cameroon"

_pathogens, 2022, doi:10.3390/pathogens11050504_

Round 1
Reviewer 1 Report
Multi-drug resistant and Extended-Spectrum Β-Lactamase (ESBL) - Producing Enterobacterales Isolated from Carriage Samples among HIV infected Women in Yaoundé, Cameroon
General comments:
Few data regarding MDR and ESBL-producing Enterobacterales isolated from vaginal swabs among WLHIV in sub-Saharan Africa have been reported so far. Therefore, I found this topic of scientific interest, this kind of surveillance is important to develop strategies to address the drug resistance problem. Still some questions need to be addressed before its acceptance for publication:
- Check the spelling to standardize the text. E.g: colonized, colonised; characterize, characterize.
- Make sure you Always write blaCTX-M-group-1,2,9 and not blaCTX-M-1,2,9. Check it in the text.
- Footnotes missing, define all the abbreviations throughout the text, including tables and figures
- The words positive and negative appear as positif and negatif in the text, please correct it.
- Check the number of MDR isolates detected, I noticed differences in the text and in the tables.
- The authors found AmpC, OXA-48 and KPC positive isolates, it would be interesting to have it discussed in the Discussion section.
ABSTRACT:
- Line 34: Don’t start a sentence with number, instead of 120 use One hundred and twenty…
- Line 41: The prevalence of vaginal carriage of MDR Enterobacterales is different from Table 1 (which is “Table 2” in the text, line 102, correct it).
- Lines 43-44: Add the number of samples out of a total to facilitate the comprehension, just like you did in this sentence “the prevalence of vaginal carriage of MDR Enterobacterales was 20.83% (22/120)”.
- Line 45: Correct 48% and 72%, use 48.0% 72.0% (check it in all the text)
INTRODUCTION
- Line 58: Comma after “communities”
- Line 81: It seems the abbreviation of ESBL-producing Enterobacterales is ESBL-PE, which you use in the discussion, add the abbreviation here so you can use it throughout the text “genotypically MDR and ESBL-producing Enterobacterales (ESBL-PE) isolated from vaginal swabs…”
- Line 62: Even defining ESBL-E in the abstract, you need to do it again when you start the manuscript
- Line 73: Rewrite the sentence to be more clear
RESULTS
- Line 88: Add the number of samples out of a total to facilitate the comprehension. “Among the 120 included WLHIV, 37 (30.83%) were colonized by Enterobacterales with 62.16% (23/37) being colonized by MDR-E. Of these women, 47.82% (11/23) also presented E-ESBL.
- Line 102: It is Table 1, not Table 2
- Line103: Replace “statues” for “status”
- Table 1: Please add “n” before the percentage sign “n (%)”
- Line 110: Rewrite the sentence, it is not clear. Add the number of samples out of a total, and then write the percentage in parentheses. Ex: 70/200 (35.0%)
- Line 114: Rewrite the title of Table 2 as follows: “Distribution of ESBL producers among MDR and non-MDR Enterobacterales isolated in WLHIV.”
- Table 2: Please, explain what did you compare to find both P-values.
- Line 131: Change “Paternity” to “Pattern”
- Figure 1: Footnotes missing, define all the abbreviations in all tables and figures
- Table 7: Check the spelling
- Line 149: It’s not necessary to write this information in the text because it is already in the table.
- Lines 97 to 101: Some information in the text doesn’t match the information on Table 2. Please, check it.
- Lines 119-120: “80.0%”, “88.0%”, “96.%”
- Line 136: Please, clarify if Table 3 is about MDR-ESBL- (MDR negative, ESBL negative), MDR+ESBL- (MDR positive, ESBL negative), or MDR+ESBL+ (MDR positive, ESBL positive). It is written MDR-ESBL-Enterobacterales, and the total number of isolates is 25; However, the number of MDR-ESBL- on Table 2 is 13. On lines 124 and 126 you wrote MDR-ESBL-positive and MDR+ESBL positive. Standardize the names throughout the text to facilitate the comprehension
- Line 145: Here we have the same problem as above, in the figure title you wrote “Resistance levels of MDR-ESBL positif and MDR- ESBL negatif Enterobacterales.” (Correct the grammar), but in the figure you have MDR+ESBL-E- and MDR+ESBL-E+. As a standard you can write MDR+ESBL positive or MDR+ESBL+ (MDR positive, ESBL positive).
- Line 146: Rewrite Table 4 title as follows: “Antimicrobial Resistance Profile of the 24 MDR Enterobacterales isolated among WLHIV.” At least on Table 4 the number of MDR isolates is 24, which is different from the Results section and other tables, please check it.
DISCUSSION
- Line 180-181: Add the percentage sign
- Lines 183-184: Rewrite the sentence as follows “where children that had an undetectable viral load (45.16%) presented the highest number of ESBL-E in fecal samples”.
- Line 187: Rewrite the sentence as follows “These rates are higher to those obtained in an Ethiopian study assessing Enterobacterales in clinical samples where E. coli accounted for 51.54% (150/291) of MDR-E isolates”
- Line 188: Rewrite the sentence to make it clear
- Line 189: I don’t think there is a discrepancy between this study and the Ethiopian study cited in the text regarding the E. coli prevalence, the numbers are very similar (56.0% x 51.54%)
- Line 191: Write the definition of ESBL-PE before use only the abbreviation
- Line 191: In the Material and Methods section the number of ESBL-E isolates is 11 and here is 12. Please, check which one is the correct.
- Line 197: Replace the sentence with the following text “These results are comparable to those obtained in a study conducted in Togo, where E. coli (63.64%) and K. pneumoniae (27.27%) were the most common ESBL-E species isolated from vaginal swabs in clinical samples
- Line 207: The word “are” is repeated
- Line 230: Rewrite the following sentence to make it clearer “…and with similar percentage of 91.66% (11/12) among MDR-ESBL-E.”
- Line 234: Replace “heathcare” for healthcare
MATERIALS AND METHODS
- The detection of the blaTEM, blaCTX-M and blaCTX-M-group-1,2,9 genes by PCR was done in which isolates (all the MDR? Only in the ESBL phenotypic positive isolates)? Please, clarify it in this section.
- It would be great if the authors could characterize the CTX-M enzyme types of groups 1, 2, and 9, and TEM (sequencing the PCR amplification products) to confirm the ESBL genotypically.
- Another suggestion is to search for blaSHV gene and characterize the enzyme in all isolates resistant to the penicillin and/or cephalosporin antibiotics families.
Author Response
please see the attachement

Reviewer 2 Report
Manuscript number: pathogens-1695525
Title: Multi-drug resistant and Extended-Spectrum Β-Lactamase (ESBL) - Producing Enterobacterales Isolated from Carriage Samples among HIV infected Women in Yaoundé, Cameroon
The present study aimed to investigate the characteristics of multidrug-resistant (resistance to at least one antibiotic of three or more families of antibiotics) as well as extended spectrum β-lactamase producing Enterobacterales (ESBL-E) isolated among clinical and asymptomatically healthy patients, especially in women living with HIV (WLHIV) from the Yaoundé Central Hospital, Yaoundé, Cameroon. They collected the large number of samples from human vagina to show the prevalence strains as well as High rates of resistance to antibiotics. Overall, the manuscript is clear and concise, and the experiments appear to have been performed conscientiously.
